# Improving Few-Shot Design Optimization By Exploiting Auxiliary Information

## Abstract

Many real-world design problems involve optimizing an expensive black-box function $f(x)$, such as hardware design or drug discovery. Bayesian Optimization has emerged as a sample-efficient framework for this problem. However, the basic setting considered by these methods is simplified compared to real-world experimental setups, where experiments often generate a wealth of useful information. We introduce a new setting where an experiment generates high-dimensional auxiliary information $h(x)$ along with the performance measure $f(x)$; moreover, a history of previously solved tasks from the same task family is available for accelerating optimization. A key challenge of our setting is learning how to represent and utilize $h(x)$ for efficiently solving *new* optimization tasks beyond the task history. We develop a novel approach for this setting based on a neural model which predicts $f(x)$ for unseen designs given a few-shot context containing observations of $h(x)$. To evaluate our method, we develop a new benchmark task involving designing customized robotic grippers for stably grasping objects. On this task, our approach which incorporates $h(x)$ significantly outperforms a baseline which only uses reward information, demonstrating improved few-shot prediction capability and more efficient optimization.

## 1 Introduction

Many real-world design problems involve optimizing an expensive black-box function $f(x)$ over a design space, such as drug discovery (Stanton et al., 2022; Gómez-Bombarelli et al., 2018), robotic hardware design (Liao et al., 2019), or hyperparameter tuning of machine learning models (Wang et al., 2024). The cost of evaluating such functions turns optimization into a problem of adaptive experimentation. Bayesian Optimization (BayesOpt) has emerged as a powerful, sample-efficient method for this problem. At a high level, BayesOpt sets up a probabilistic surrogate model $G$ of the objective function $f$. At each iteration of optimization, this model is used to guide an acquisition function to select a point $\mathbf{x}$ of interest, subsequently $f(\mathbf{x})$ is queried, and $G$ is updated with the resulting feedback, repeating until termination when the best design $f(\mathbf{x}^*)$ is returned.

While these methods have proven useful within the basic black-box setting, the setting itself is highly restrictive in comparison to modern scientific or engineering experimental setups. In many real-world problems, the experimenter has access to fine-grained observations of the system, or the capacity to obtain multiple performance measures correlated with the ultimate metric of interest. We might posit such a setting as one where the experimenter receives "extra information" beyond a single scalar reward, which could help guide subsequent experimentation. For example, in robotic hardware design, a real-world trial of a robot may generate a time series of sensor observations along with a final performance measure of the design (e.g., grasp stability). Such extra information is presumably useful for understanding *how* one design succeeds while another fails, and could be altered to succeed.

Concretely, we posit an optimization setting where evaluating $\mathbf{x}$ returns both a reward $f(\mathbf{x})$ and some auxiliary (potentially high-dimensional) information $h(\mathbf{x})$, which is presumably correlated with $f(\mathbf{x})$ and useful for optimization. A common example is when evaluating a design yields a sequence of observations over time $O_1, .... O_t$ correlated with $f(\mathbf{x})$, such as in the robotics design example above. We additionally posit that the (automated) experimenter has access to a history of related tasks $T_1, ... T_n$ with a shared form of the auxiliary information $h$, relevant to solving the

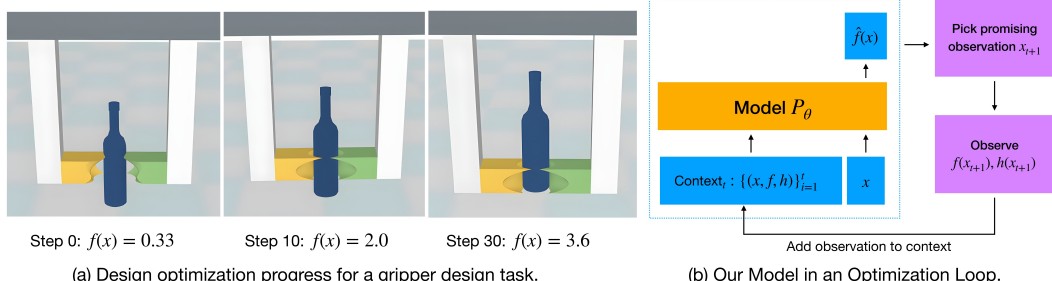

Step 0: $f(x) = 0.33$   Step 10: $f(x) = 2.0$   Step 30: $f(x) = 3.6$

(a) Design optimization progress for a gripper design task.       (b) Our Model in an Optimization Loop.

Figure 1: **Few-shot design optimization with our method**. On the left, we show our design method applied to a gripper design task, where the goal is to design a stable, perturbation-resistant gripper for an object given auxiliary tactile information during a grasping trial. Our method converges from a simple friction grasp in trial 0 to a stable, high-reward gripper by trial 30, successfully utilizing the auxiliary information $h(\mathbf{x})$ provided by each trial for efficient optimization. On the right, we show our method in an optimization loop. We train a model $P_\theta$ for probabilistic prediction of $f(\mathbf{x})$ for an unseen design $\mathbf{x}$, given a few-shot context of observed designs including $h(\mathbf{x})$ for each observation. The model's predictions are used to select a promising next design $\mathbf{x}_{t+1}$ for evaluation, the design is evaluated, and the feedback $f, h$ for the design is added into the context, repeating until termination.

current task. This is similar to a human experimenter who develops intuition about what to observe in an experiment through previous experience, where the form of observations $h$ is most often similar across tasks (e.g. dynamics of a physical system). It is also a common feature in many real-world optimization problems.

Prior work has examined a simpler version of this setting by assuming a single-task with composite structure $f(\mathbf{x}) = g(h(\mathbf{x}))$ and observations of $h$ along with $f$ (Astudillo & Frazier, 2019). Our setting does not strictly assume $h$ arrives from a composite structure, which restricts the form of $h$ and excludes potentially rich auxiliary information correlated with $f$. Moreover, we assume that a task history is available with observations of $h$. Crucially and different from prior work, this requires a design method to generalizably *learn* how to represent the auxiliary information from the task history, such that it can efficiently optimize new tasks. This can be highly useful; since $h(\mathbf{x})$ provides rich information about the task, utilizing it successfully can enable deeper structural understanding and more systematic, efficient search for high-quality designs. However, making use of this information, which could be high-dimensional and heterogeneous, in order to solve unseen design tasks is a difficult challenge. Notably, it shifts black-box optimization partially to a representation learning question, of how to encode $h(\mathbf{x})$ in a way that offers insight for new tasks.

We introduce a novel approach for this new setting. Our approach is to learn a neural model to predict the performance of unseen designs given a small context of evaluated designs for a task. Importantly, the auxiliary information is made available in the context, and our method learns how to makes use of $h(\mathbf{x})$ while predicting $f(\mathbf{x})$ for unseen designs. We adopt a transformer-based architecture for our model, which recent work has shown to be effective for few-shot uncertainty-aware prediction (Nguyen & Grover, 2022). The model is trained on a set of tasks with existing evaluations, and applied as a surrogate model for optimization of a new task (Fig. 1).

We create a new design task for benchmarking our proposed method, based on designing customized robotic grippers for objects using tactile feedback over the course of a grasp (Figure 1). This task represents a challenging benchmark for design optimization methods where $h(\mathbf{x})$ is high-dimensional and learning its utility for optimization is highly non-trivial. On this task, our method significantly outperforms a baseline that does not exploit auxiliary information, demonstrating improved few-shot prediction capability and more sample-efficient optimization.

In sum, our work makes three key contributions: (1) a new design optimization setting with 'auxiliary information' available with multiple tasks, offering fundamental new challenges, (2) a novel method for this task based on a neural model for auxiliary information-aware few-shot prediction, and (3) a new gripper design task as a benchmark for this setting, where we demonstrate our method's improvements on prediction and optimization. Overall, our work takes a step towards more capable systems for optimization and experimental design, which can operate effectively in more realistic, high-information environments for design and discovery.

## 2 BACKGROUND AND RELATED WORK

**Background on Bayesian Optimization.** Bayesian Optimization (BayesOpt) is a framework for optimization of expensive black-box functions. BayesOpt maintains a surrogate model $M_\theta$ of the function $f(\mathbf{x})$, which outputs a probabilistic prediction $M_\theta(\cdot|\mathbf{x})$ of $f$ at any design point. Given observations of $f$, the surrogate model updates its initial prior $P(f)$ to a posterior $P(f|\mathcal{D})$. A popular choice for $M_\theta$ is a Gaussian process (GP), which produces an analytic posterior $M_\theta(\cdot|\mathbf{x}, \mathcal{D}) = \mathcal{N}(\mu_{\mathbf{x}|\mathcal{D}}, \sigma_{\mathbf{x}|\mathcal{D}})$. Neural models such as BNNs (Li et al., 2023) are also common choices.

The second component of BayesOpt is an acquisition function $\alpha(\mathbf{x})$, which quantifies the value of observing $\mathbf{x}$ based on the surrogate's predictions $M_\theta(\cdot|\mathbf{x}, \mathcal{D})$. A common acquisition function is Probability of Improvement: $\alpha_{\text{PI}}(\mathbf{x}) = \mathbb{P}[\hat{f}(\mathbf{x}) > f_{\text{best}}]$, i.e. the probability that observing at the current point yields improvement over the currently best value. At each iteration, a new observation point is selected by solving the optimization problem $\mathbf{x}_t = \arg\max_{x \in \mathcal{X}} \alpha(\mathbf{x})$. Subsequently $f(\mathbf{x}_t)$ is added to $\mathcal{D}$ and $M_\theta$ is updated. At termination, the best value $f(\mathbf{x}^*)$ is returned.

As an adaptive experimentation method, BayesOpt has many applications to design, when the system's dynamics are unknown or costly to simulate, and $f$ must be treated as a black-box. For example, drug design involves assessing difficult-to-simulate interactions where exhaustive search of a molecule library is impossible Stanton et al. (2022). Hardware design, such as for robotics or chip design, is another application where expensive real-world trials are required. In these applications, it is reasonable to expect that experimentation yields a wealth of useful information.

**Composite Bayesian Optimization**. Extending Bayesian Optimization to include additional information has primarily been studied under the *composite* setting, where it is assumed that $f(\mathbf{x}) = g(h(\mathbf{x}))$ and $h$ is observed along with $f$. Astudillo & Frazier (2019) first studied this setting, and proposed a GP on $h$ with a Monte-carlo approximation of Expected Improvement (EI) as the acquisition function. Subsequently, this work has been extended by assuming specific forms of $h$ such as intermediate values of nested functions Astudillo & Frazier (2021), or to higher-dimensional versions of $h$ via a neural network that maps to lower dimensions Maus et al. (2023). All these works assume a single-task optimization setting, where any learning is contained within a task and significantly limited by the need for sample-efficient optimization. In our setting, $h$ is assumed to be observed in a history of multiple tasks, and a key challenge is to learn a robust representation of $h$ that is generalizable to a completely *new* task at test-time. Note also that previous work models the composite information with GPs, which have stricter assumptions and high computational cost, limiting the complexity of representations achievable compared to our neural model-based approach.

**Transfer/multi-task Bayesian Optimization.** Several works have explored the transfer setting in BayesOpt, where a dataset of 'training' tasks with functions $f^{(1)}, ..., f^{(n)}$ are available to accelerate optimization of a new test task with function $f_{\text{test}}$. For each training task, there is an associated dataset of function evaluations $D^{(i)} = \{(\mathbf{x}_j^{(i)}, f^{(i)}(\mathbf{x}_j^{(i)}))\}_{j=1}^{n_i}$. Approaches for this case divide into putting all data-points into a single surrogate like a multi-task GP (Swersky et al., 2013), learning a mean and kernel prior from training tasks (Wistuba & Grabocka, 2021; Wang et al., 2024), ensembling single-task models Wistuba et al. (2018), or learning an acquisition function to transfer to new tasks (Volpp et al., 2019). Importantly, the methods in this section do not make use of auxiliary information, and generally capture a logic of 'matching' a test task to the most similar training task.

**Neural Networks for Black-Box Optimization**. Neural networks were first applied to BayesOpt methods for learning the kernel function in GP models (Wilson et al., 2016; Wistuba & Grabocka, 2021; Wang et al., 2024), or less commonly with Bayesian NNs as a surrogate (Snoek et al., 2015; Li et al., 2023). More recently, neural processes (NPs) were proposed as methods that learn the predictive posterior directly from data, by predicting the value of $f(\mathbf{x})$ for a 'target' set from a small 'context' of observations (Garnelo et al., 2018a;b). Recent variants of NPs with transformer-based architectures have shown strong predictive abilities (Nguyen & Grover, 2022; Müller et al., 2021). A final class of models take an end-to-end approach and directly output the next point to acquire from the observation history (Chen et al., 2022; Maraval et al., 2023), although this requires optimization trajectories at training time or sample-inefficient RL approaches. Our setting challenges models to incorporate auxiliary information in a generalizable way, and thus presents a more challenging learning task that requires a novel approach. We introduce our approach and model in Sec. 4.

## 3 PROBLEM SETTING

We formalize the problem of black-box optimization with auxiliary information as follows. We consider a task $\mathcal{T}$ with associated function $f(\mathbf{x})$ over a design space $\mathcal{X}$, and aim to solve the maximization problem $\max_{\mathbf{x} \in \mathcal{X}} f(\mathbf{x})$. Different from the basic setting, we assume that a trial generates auxiliary information $h(\mathbf{x})$ along with $f(\mathbf{x})$, which could be high-dimensional. We thus define a trial as a vector-valued function $F(\mathbf{x}) = (f(\mathbf{x}), h(\mathbf{x}))$, leading to the following rephrased problem:

$$\mathbf{x}^* \in \arg\max_{\mathbf{x} \in \mathcal{X}} F(\mathbf{x})_0, \tag{1}$$

where $F(\mathbf{x})_0$ refers to the first element of $F(\mathbf{x})$. Rather than starting to optimize a 'target' task $\mathcal{T}$ *tabula rasa*, we consider the multi-task/transfer setting where there is a dataset of related 'training' tasks $\mathcal{T}^{(1)}, ... \mathcal{T}^{(N)}$, each associated with functions $F^{(1)}, ... F^{(n)}$. For each source task $\mathcal{T}^{(i)}$, the dataset contains a set of evaluations $\mathcal{D}^{(i)} = \{(\mathbf{x}_j^{(i)}, F^{(i)}(\mathbf{x}_j^{(i)}))\}_{j=1}^{n^{(i)}}$, where $n^{(i)}$ is the number of evaluations for task $\mathcal{T}^{(i)}$. These tasks can be used as (pre)training data to learn generalizable features of the task family, applicable to optimizing a new task at test-time, i.e. solving Equation 1.

*Practical Applications.* Many practical problems are encompassed by this setting. In robotics, different robot tasks may require custom hardware or specific tools customized for robot morphology, while a trial of a hardware/tool configuration generates high-dimensional sensor observations Liao et al. (2019). In drug design, optimizing binding to a target of interest involves *in-vitro* experiments that can measure multiple metrics/attributes of the candidate molecule, while benefiting from experience with related targets (Ramakrishnan et al., 2014). Even hyperparameter tuning of large neural networks may benefit from full observations of loss curve behaviors (Adriaensen et al., 2023).

*Problem Features and Methodological Questions.* As discussed in above sections, this setting presents interesting methodological questions beyond the basic setting. Most significantly, the presence of the auxiliary information $h(\mathbf{x})$, and a history of related tasks involving the same form of $h$, raises the question of how to build a useful representation of $h$ across a range of tasks. Intuitively, while $f(\mathbf{x})$ states the fact of a design's quality or lack thereof, $h(\mathbf{x})$ could more precisely suggest *how* one design fails while another one succeeds.

Thus, a crucial feature of this setting is that the design method must *learn* the 'deeper' information present in $h(\mathbf{x})$ in a generalizable way, applicable to solving entirely new optimization tasks. This learning is enabled by the task history, but presents a challenging task as $h(\mathbf{x})$ may be complex and high-dimensional, while having a nontrivial relationship with the objective $f(\mathbf{x})$. We note that reasoning about the objective by way of similarity in input space (i.e. $||\mathbf{x} - \mathbf{x}'|| \implies |f(\mathbf{x}) - f'(\mathbf{x})|$) is still essential, as captured by previous methods; however, an equally essential aspect is how to encode $h(\mathbf{x})$. This also indicates the key test of a method for this setting: to show that utilizing $h(\mathbf{x})$ outperforms any baseline that makes use of the objective function alone, on unseen test tasks.

## 4 METHOD

We propose a novel method for the setting inrtoduced in Section 3. Our method is based on a neural model which learns to perform few-shot probabilistic prediction. Concretely, given a small context of observed designs $\{\mathbf{x}\}_C$ for a task, in which *both* $f(x)$ and the auxiliary information $h(x)$ are observed, the model produces a probabilistic prediction for the performance $f(\{\mathbf{x}\}_T)$ for a set of unseen designs. In Sec. 4.1, we detail our foundational approach and the learning problem, agnostic to the particular choice of neural model. In Sec. 4.2, we describe our transformer-based architecture for this few-shot prediction model. Finally, in Sec. 4.3 we discuss how our model can be used in a Bayesian Optimization procedure.

### 4.1 FOUNDATIONAL APPROACH

In our setting, we associate a task $\mathcal{T}$ with an objective $f(\mathbf{x})$ and auxiliary information function $h(\mathbf{x})$. As mentioned in Sec. 3, we consider a trial as a vector-valued function $F(\mathbf{x}) = (f(\mathbf{x}), h(\mathbf{x}))$, which returns both outputs. Since $f(\mathbf{x})$ and $h(\mathbf{x})$ are correlated, we treat the addition of auxiliary information by taking the viewpoint of a prior $P(F)$ over the joint function, and consider a given

task as a sample from this prior $F_i \sim P(F)$. Conditioning this prior on a set of observations $\mathcal{D} = \{(\mathbf{x}_i, F(\mathbf{x}_i))\}_{i=1}^{n}$ produces a predictive posterior $P(F|\mathcal{D})$. From this viewpoint, we seek a model $P_\theta$ that can approximate the true prior $P(F)$, or equivalently, approximate the predictive posterior distribution such that $P_\theta(F|\mathcal{D}) \approx P(F|\mathcal{D})$.

While $P(F|\mathcal{D})$ could be approximated via a manually specified prior (e.g. a Gaussian Process), this could be conceptually limiting ($h(\mathbf{x})$ may have a complex relationship with both $f(\mathbf{x})$ and $\mathbf{x}$), and computationally intractable ($h(\mathbf{x})$ could be very high-dimensional, e.g. a time-series). Instead, we *learn* a neural network to approximate the predictive posterior distribution (PPD) of $F(\mathbf{x})$ on a finite dataset, allowing it to learn a useful representation of $h(\mathbf{x})$ in the process. In the basic formulation, the neural network accepts a context set $C = \{(\mathbf{x}_i, F(\mathbf{x}_i))\}_{i=1}^{n_C}$ and target set $T = \{\mathbf{x}_i\}_{i=1}^{n_T}$, and produces a prediction $P_\theta(\cdot|C, T)$ of the value of $F$ for the target inputs. It learns this few-shot prediction ability from a large set of training tasks, each assumed to be sampled from $P(F)$.

A challenge with this basic formulation is that $h(\mathbf{x})$ can be very high-dimensional and potentially heterogeneous. Apart from the obvious computational challenges of predicting $h$, there is a more fundamental issue: while $h(\mathbf{x})$ is correlated with $f(\mathbf{x})$, predicting the full content of $h(\mathbf{x})$ may be misaligned with the goal of modeling and optimizing $f(\mathbf{x})$. For example, if $h$ is a time sequence of observations, only a portion of this sequence might actually be valuable for identifying promising new designs. Accordingly, the model's learning objective should focus on utilizing $h(\mathbf{x})$ for predicting the *performance* of new designs, rather than requiring full reproduction of $h$. Thus, our approach instead models the PPD $P_\theta(f(T)|C, T)$, where the context still contains evaluations of $F(\mathbf{x})$.

We thus set up the following few-shot prediction task for the model at training time. The training dataset consists of tasks $\mathcal{T}^{(1)}, ..., \mathcal{T}^{(n)}$ from a common distribution. At each iteration, a task $\mathcal{T}^{(i)}$ is sampled uniformly, and non-overlapping subsets of its dataset $D^{(i)}$ are allocated to a context set $C = \{(\mathbf{x}_j^{(i)}, F(\mathbf{x}_j)^{(i)})\}_{j=1}^{n_C}$ and a target set $T = \{\mathbf{x}_j^{(i)}\}_{j=1}^{n_T}$. Note that this procedure induces a distribution $Q(C, T)$ over context and target sets. The model makes a probabilistic prediction $f(T)$ for the target inputs, and minimizes the following objective:

$$L(\theta) = \mathbb{E}_{C, T \sim Q}[-\log P_\theta(f(T)|T, C)]. \tag{2}$$

For simplicity, we assume that the distribution over target function values can be factorized given the context, i.e. $P_\theta(f(T)|T, C) = \prod_{\mathbf{x}_i \in T} P_\theta(f(\mathbf{x}_i)|\mathbf{x}_i, C)$. We do not expect that this assumption is too restrictive in practice, but note that it could also be relaxed if necessary. For each target input, the model predicts a univariate normal distribution for the value of $f$. In practice, we sample a batch of tasks at each iteration, and average the likelihood across tasks when computing the final loss.

## 4.2 Model Architecture

In this section, we discuss our transformer-based architecture for realizing the model $P_\theta$ discussed above. Transformers have several attractive properties for this task: they can produce predictions invariant to the order of context and target sets by choosing a suitable attention structure. Moreover, attention mechanisms are well-suited to associating a target input $\mathbf{x}$ with relevant context observations of $F$ that enable accurate prediction of its performance.

Our architecture is shown in Figure 2. Each data-point in the context and target sets is considered an individual token for the model. Each token is encoded to a fixed dimension and then passed into the transformer. Finally, a small MLP is applied to the last-layer target token representations to predict the function value as a distribution $\mathcal{N}(\hat{\mu}, \hat{\sigma})$. As first proposed in Transformer Neural Processes Nguyen & Grover (2022), we remove positional encodings and use an attention structure where the context points attend to each other, while each target point attends only to the context points.

A crucial difference from prior work is the presence of $h(\mathbf{x})$ in the context, which could be high-dimensional. Therefore, the input encoder requires careful design to ensure that $h(\mathbf{x})$ is represented effectively and captures the full wealth of information, while not also diluting the influence of $\mathbf{x}$ and $f(\mathbf{x})$. We address this by first adopting separate input encoders $E_\theta^c((\mathbf{x}, f, h))$ and $E_\theta^t(\mathbf{x})$ for the context tokens and target tokens. While the design of the context encoder depends on the particular form of $h(\mathbf{x})$, we focus our discussion on a common case when $h(\mathbf{x})$ is a time-series of observations $O_1, ...O_T$. The architecture of this input encoder is shown in Fig. 2b. We encode $h$ itself using

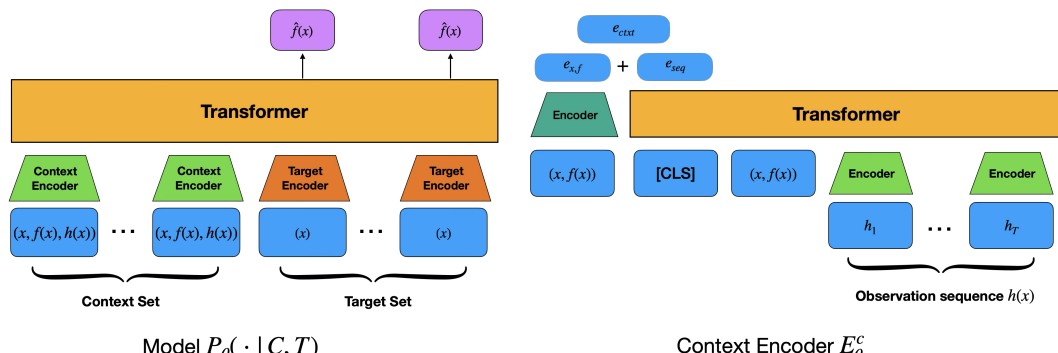

Figure 2: **Model architecture for our method.** On the left shows our model for few-shot probabilistic prediction. A small context set of observations is provided as conditioning, in which observations of $f(\mathbf{x})$ and $h(\mathbf{x})$ are available. A target set with only inputs $\mathbf{x}$ is also provided, encoded separately. The encoded data-points pass through a transformer where the target points can only attend to the context, and a prediction of $f$ is made for each of the target points. The right shows the form of the context encoder when $h(\mathbf{x})$ is an observation sequence. A transformer encoder embeds the encoded sequence, which is added to an embedding of $(\mathbf{x}, f(\mathbf{x}))$ to obtain the final embedding $e_{ctxt}$.

a transformer encoder, adding a [CLS] token to obtain a sequence embedding. Note that a special token encoding $(\mathbf{x}, f(\mathbf{x}))$ is also added, since the valuable parts of the sequence to attend to may depend on this information. Finally, we obtain the embedding for a context observation by adding together the sequence embedding and an embedding of $(\mathbf{x}, f(\mathbf{x}))$.

### 4.3 USING OUR METHOD FOR BAYESIAN OPTIMIZATION

Once our model $P_\theta$ is trained, we can use this model as a surrogate model in a Bayesian Optimization loop. For a given test task $\mathcal{T}$, an initial set of observations is collected as a small context $C_0 = \{\mathbf{x}_i, F(\mathbf{x}_i)\}_{i=1}^{n_0}$. At each iteration $t$, the next observation is selected by solving the optimization problem $\mathbf{x}_{t+1} = \arg\max_{\mathbf{x} \in \mathcal{X}} \alpha(P_\theta(\cdot|\mathbf{x}, C))$. Subsequently, $F(\mathbf{x}_{t+1})$ is observed and the result is appended to the context to form $C_{t+1}$, repeating until termination. Note that the model itself is not updated during optimization, presenting an advantage over methods that require iterative re-training.

In our experiments, we assume that $\mathcal{X}$ is discrete and optimization of $\alpha$ proceeds over a finite (potentially large) set of designs. However, we note that our method could also be applied to continuous input spaces, by differentiating $P_\theta$ w.r.t. the target input and optimizing $\alpha$ via gradient ascent.

## 5 EXPERIMENTAL SETUP

To evaluate methods for our setting, we develop a new black-box design task, where the aim is to design the shape of robotic grippers using tactile feedback. Many important applications require task-specific grippers customized for grasping specific objects, such as industrial applications or automated scientific experimentation (Burger et al., 2020). Recent works have consequently studied the problem of generating customized gripper designs automatically for different objects (Ha et al., 2021; Xu et al., 2024). We study a variant of the gripper design problem where the gripper must be designed via tactile feedback. This has connections to the "blind grasping" problem in robotics, where a robot must grasp an object via tactile feedback alone (an ability that comes naturally for humans) (Dang et al., 2011).

Below, we describe our novel benchmark task. Note that this task fits the desiderata of our setting: (1) several trials may be required to find a high-performing solution, (2) the tactile information ($h$) obtained through evaluation of a design could be useful for refining the design, and (3) experience designing grippers for many objects should transfer to a novel test object. Thus, this task serves as a useful benchmark for design methods, and we expect it to be of value to researchers in the future.

**Task Definition and Simulation Details**. We define the gripper design task as follows: a task $\mathcal{T}$ involves finding a gripper geometry $\mathbf{x}$ that can grasp an object $O$ as stably as possible. The geometry parameter vector $\mathbf{x}$ is used symmetrically to form two gripper fingers in a parallel-jaw gripper. To

evaluate the gripper quality, we run a simulation which measures grasp stability by perturbing the object with disturbance forces (the most common way to evaluate grasps (Roa & Suárez, 2015)).

*Simulation Details.* The simulation setup is shown in Fig. 3. The parallel-jaw gripper consists of a wrist with left and right handles, to which the fingers are attached. The object starts on the ground centered between the gripper fingers. We parametrize the gripper geometry as a cubic Bezier surface, a flexible and smooth surface parametrization. The shape of the surface is controlled by 16 3-dim control points; the $y$-coordinate of each control point is optimized along with the $x$ and $z$ coordinates of the interior control points. The surface is extruded to 3D to form gripper fingers. We additionally optimize the initial *height* of the gripper as a policy parameter, for a 21-dimensional design space.

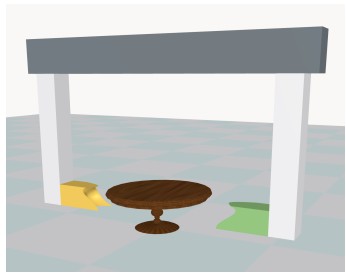

Figure 3: Simulation at $t = 0$.

The simulation proceeds as follows. The gripper fingers close in on the object with $1.0$N force along the $y$-axis, and lift up the object to a fixed height (or fail to lift). Then, disturbance forces are applied to the object in a fixed direction, with increasing magnitude. We run three simulations with different force directions to test robustness of the grasp to different perturbations: $-z$ (down), $-x$ (into page), and $+x$ (out of page). For each simulation, we start applying a force of $F = 0.5$N and increment the force by $0.1$N while the object remains in the grasp (waiting after each application for the object to re-stabilize). The reward is the maximum force applied before the object falls out of the grasp; the final reward is the average across the three simulations. We run the simulation for maximum 5000 steps, with a maximum possible reward of $\sim 6.0$. We implement the simulation in MuJoCo (Todorov et al. (2012)).

*Tactile feedback.* In addition to the reward, each simulation produces a tactile measurement at each time-step. We divide the inward face of each gripper into a $16 \times 16$ grid of "taxels". If a contact point falls within a particular taxel, the resulting contact force is registered within that taxel. This yields two $16 \times 16$ tactile images $\text{Ti}_L$ and $\text{Ti}_R$ for the left and right grippers. We also record scalar contact readings for the other gripper faces and attached handle, yielding 5-dim readings $\text{Tf}_L$ and $\text{Tf}_R$. These tactile readings at each time-step, along with minimal state information (gripper position, velocity, etc.), compose the auxiliary information $h(x)$. Further details can be found in the Appendix.

**Dataset Generation and Statistics**. In order to robustly learn a few-shot predictor that incorporates auxiliary information, we generate a large dataset of tasks. To do so, we leverage the ShapeNet dataset (Chang et al., 2015), a rich collection of 3D models for many common objects (e.g. tables, cars, sofas). We sample $\sim 1$k objects from ShapeNet for our dataset. Each object is normalized to fit in a $(9\,\text{cm})^3$ box and placed on the ground between the gripper fingers (of size $(2\,\text{cm})^3$). For each object, 400 (unique) random Bezier geometries are generated, and each gripper geometry is evaluated at height increments of 0.5 cm until the height of the object is exceeded (for a range of 800-7200 evaluations). These evaluations compose the dataset $\mathcal{D}^{(i)}$ for each of the training tasks.

The final dataset consists of 4.28 million designs evaluated across 997 unique objects. This is a large-scale dataset where a model could learn rich representations for optimization. We note that most of the tasks have good solutions; 31.3% of objects have a gripper design that achieves the maximum possible reward (6.0), and 65.3% have a design with at least $83.3\%$ of the max reward. In general, high-performing designs are more sparse, making them challenging to discover. We divide the dataset into a test set (150 objects), a validation set (75 objects), and a training set (772 objects).

## 6 RESULTS

**Implementation Details**. Our model $P_\theta$ (Fig. 2) is trained on the gripper design task (Sec. 5) by sampling a training task, and performing few-shot prediction of grasp stability from a context which includes tactile information as $h(\mathbf{x})$. We sample context lengths uniformly in the range $[5, 30]$ for training, and the target set size is fixed at 100 examples. For both sets, we sample low-reward and high-reward data-points with equal probability, ensuring that the model has an 'informative' context and also learns to accurately predict high-performing designs. We use a model dimension of 256 and 9 layers for both the context encoder transformer and the predictor transformer. Each time-step of the tactile sequence $h_t$ is encoded with a small CNN followed by an MLP; see Appendix for details.

The target encoder is a one-hidden-layer MLP. The '$f$-only' baseline only provides the value of $f(\mathbf{x})$ in the observation context, and uses MLPs for the context and target encoders. We train both models with AdamW with $\eta =$1e-4 and weight decay of 0.01, dropout of 0.2, and early stopping.

## 6.1 PREDICTION RESULTS

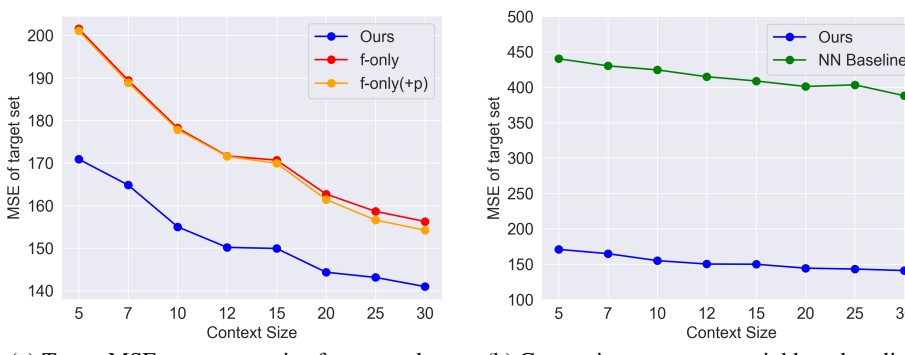

(a) Target MSE vs. context size for test tasks.  (b) Comparison to nearest neighbors baseline.

Figure 4: **Few-shot prediction results**. The left shows the MSE on the target set for different context sizes, averaged across the test tasks. Our model that incorporates auxiliary information significantly outperforms the $f$-only baseline and a variant with more parameters ($f$-only+p) on all context sizes, especially notably on smaller context sizes. On the right, we show that a baseline predicting the value of the nearest-neighbor in the context set performs poorly. For each context size, 10 different contexts were sampled for each test task and the results averaged across contexts. The reported MSE is summed over the target set size, which is 100 examples.

We first examine our model's capabilities for few-shot prediction. We focus primarily on mean squared error (MSE) as the metric (summed over the 100 target set examples), since we are focused on achieving accurate prediction of the performance measure for unseen designs. Figure 4 shows our results, averaged over the test tasks. Our model which incorporates auxiliary tactile information significantly outperforms a baseline which only incorporates $f(\mathbf{x})$. Our improvement is especially notable for smaller context sizes; for a context of 5 observations, our model achieves an MSE of 170.9 vs. an MSE of 201.6 for the baseline. This indicates that when the model has observed a smaller number of designs for a task, the tactile information is especially useful and enables the model to better understand the task and predict unseen designs. For both methods, intuitively the error decreases as context size increases. Note that our model with a context size of *10* observations has a lower error than the $f$-only model with a context of *30* observations.

Our model has more parameters than than the $f$-only model (15.1M vs. 5.5M), due solely to the $h(\mathbf{x})$ encoder. To disentangle the effect of parameter size, we include a version of the $f$-only model with more parameters than our model ($f$-only(+p), 16.7M parameters). The $f$-only(+p) model has virtually identical predictive performance to $f$-only, indicating that our model's improved few-shot prediction capabilities are due to successfully utilizing tactile information rather than model capacity. Finally, we compare to a nearest-neighbors baseline which predicts the value of the closest context input for each target input (Fig. 4b). Our method substantially outperforms this baseline, which has high MSE ($\sim 400$) indicating that this task cannot be solved by simple reasoning.

## 6.2 OPTIMIZATION RESULTS

We next evaluate our model on Bayesian Optimization of unseen design tasks. For all experiments, we run discrete BayesOpt where the model must choose observations from a finite, large set of evaluated designs (average size of 4.3K). We perform 5 runs for each test task with different initial contexts. For each run we use an initial context of 5 designs, sampled such that no design has performance more than 30% of the maximum, and run optimization for 30 trials. We use the Probability of Improvement acquisition function for all experiments.

Results are shown in Fig. 5. Our model achieves consistently higher reward across the course of optimization (Fig. 5a), and lower average regret (Fig. 5b). Improvement is particularly notable

when the number of trials is quite small, indicating that our model can exploit the richness of the auxiliary tactile information to more efficiently identify higher-performing designs. After 30 trials, on average our model reaches almost 90% of the maximum achievable reward.

We also examine the percentage of tasks where the optimization achieves the maximum possible reward for the task ($f(\mathbf{x}^*) = f_{max}$). By 30 trials, our model reaches this for 34.4% of test tasks, vs. 26.7% for the $f$-only model. With a more conservative measure of solving a task (regret $\leq$ 0.5), our model solves 67.2% of test tasks vs. 58% for the baseline (Fig. 5c shows progress over optimization). This indicates that our model is able to discover close-to or exactly optimal solutions for novel design tasks, leveraging the feedback of $h(\mathbf{x})$ in these tasks for more efficient optimization.

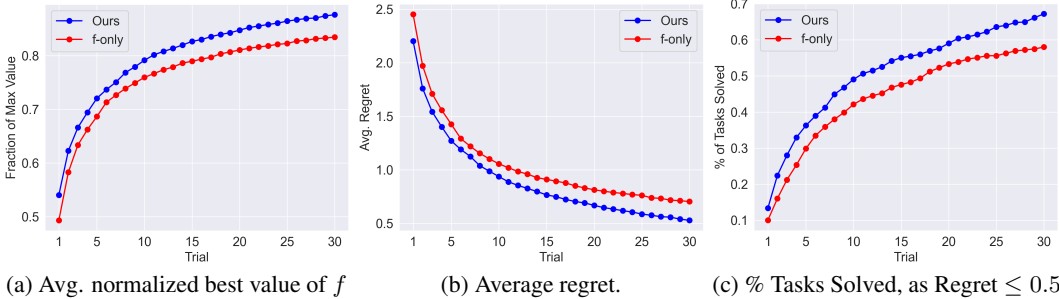

(a) Avg. normalized best value of $f$      (b) Average regret.      (c) % Tasks Solved, as Regret $\leq$ 0.5

Figure 5: **Optimization on Unseen Tasks**. Our model significantly outperforms the $f$-only model for optimization of test tasks. (a) shows the best achieved value of $f$ (normalized by max $f$ for the task) over optimization, averaged across test tasks. Our model makes substantially faster optimization progress early on by exploiting auxiliary information, and remains consistently better across optimization. (b) shows the average regret, with the same conclusions. In (c), we show that our method solves a significantly larger % of tasks than the baseline at any point in the optimization.

*Qualitative Examples.* Figure 6 shows examples of two design tasks in the test set. In the first example, at step 0, the gripper picks up a bottle by a flat *friction* grasp, which cannot withstand much disturbance. By the end of optimization, our method discovers a stable grasp that curves inward to support the sides, while also supporting the base. By contrast, the $f$-only model is unable to improve upon the initial context; presumably, our method can leverage the rich tactile feedback available even in *failures* to infer an appropriate gripper. For the case of an airplane object, by Step 30, our model finds a stable grasp that on first contact rotates the plane, pushing one wing in front and one wing behind the gripper. This suggests that the model learns to leverage *dynamics* for stable grasps. In contrast, the $f$-only model's final grasp can be dislodged by applying a force to the nose. These examples indicate our model's potential for finding creative solutions to design problems.

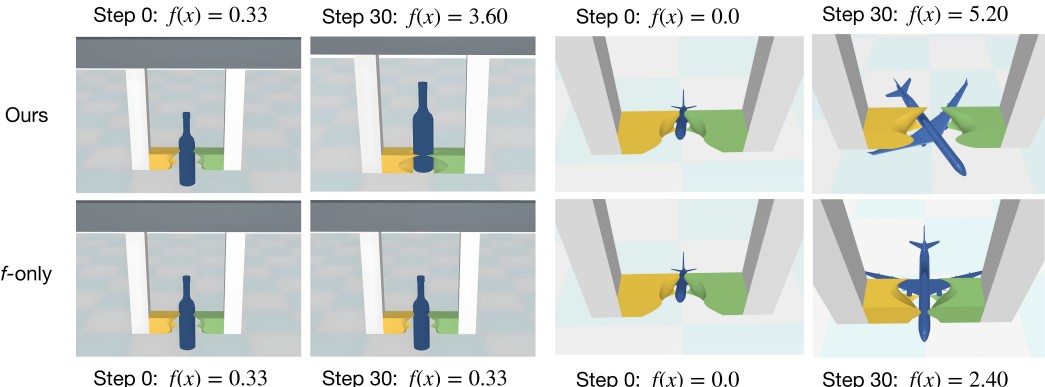

Figure 6: **Examples of design optimization.** For the left task, our model finds a stable grasp that exploits the shape of the bottle, in contrast to the $f$-only model. On the right (top-down view), the initial grasp does not pick up the airplane. Our model discovers a solution with maximum possible reward that pushes the wings to opposite sides, while the $f$-only model's solution is less stable.

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

## A APPENDIX

### A.1 GRIPPER DESIGN TASK DETAILS

We provide further details of the gripper design task below.

**Simulation.** The gripper simulation consists of a wrist with an up-down position-based actuator, and left and right handles with force-based actuators. The left and right grippers close in on the object with 1.0N of force for 250 time-steps, at which point it has settled into a grasping pattern. Then the object is lifted to a height of 0.12 m. Once the object has stabilized, disturbance forces are applied in a fixed direction starting at $F = 0.5$N and incrementing by 0.1N. Each force is applied for 20 time-steps; after each force application, the object is allowed to re-stabilize before incrementing the force. The simulation runs for a maximum of 5000 time-steps, terminating early if the object falls from the grasp and touches the ground. We use the default time-step of 0.002 seconds in MuJoCo. The object has a fixed mass of 0.05 kg, and each gripper has a fixed mass of 0.1 kg. For contact

modeling which requires convex geometries, we perform convex decomposition of both the object and the gripper using the CoACD library before running the simulation (Wei et al., 2022).

We run three simulations with different force directions: $(-z)$ (down), $(-x)$ (into the page), and $(+x)$ (out of the page). The reward $f(\mathbf{x})$ is the maximum disturbance force applied before the object fell out of the grasp, or at the end of the simulation (maximum achievable reward is around 6.0). This indicates how well the grasp was able to resist disturbances in a particular direction. The final reward for a design is this metric averaged over the three simulations $(f_1(\mathbf{x}) + f_2(\mathbf{x}) + f_3(\mathbf{x}))/3$, as a measure of robustness to perturbations in different directions.

**Tactile Feedback.** We divide the inward face of each gripper into a $16 \times 16$ uniform tactile grid. At each time-step, MuJoCo computes contact points $(\mathbf{p}_i, \mathbf{n}_i, \mathbf{F}_i)$, where $\mathbf{p}_i$ is the position of the contact, $\mathbf{n}_i$ is the contact normal, and $\mathbf{F}_i$ is the contact force in the local frame. Like standard approaches (e.g. MuJoCo's touch sensor), we record the contact force in the normal direction. We map the contact point to a 'taxel' in the grid based on its location (this often requires projecting the point to the gripper surface via the contact normal). If the contact point is on another face than the inward face (e.g. the gripper is supporting the object from its top face), a scalar contact reading is recorded for that face.

The result of this process is the following information at each time-step: two $16 \times 16$ tactile images $\text{Ti}_L$ and $\text{Ti}_R$ for the inward face of each gripper, and two 5-dimensional vectors $\text{Tf}_L$ and $\text{Tf}_R$ for the scalar contact readings on the other faces (top face, bottom face, back face, front face, and handle). Finally, we include a small amount of state information $q$ at each time-step: the position and velocity of all the joints (6-dim), the control policy for lifting the gripper (1-dim), the disturbance force applied at that time-step, which could be 0 (2-dim, $y$-component always 0), and the current time-step of the disturbance force which is applied for 20 time-steps (1-dim, 0 if no force at that time-step). The two tactile images, two face readings, and system state becomes the information at each time-step. This information is recorded at each time-step in the sequence. Since we have three simulations, we record the observation sequence for each simulation and $h(\mathbf{x})$ contains the three sequences $h(\mathbf{x}) = (h_1(\mathbf{x}), h_2(\mathbf{x}), h_3(\mathbf{x}))$. Notably, the features at each time-step are heterogeneous, and the tactile information at each time-step tends to be quite sparse; thus capturing the insights present in this complex observation sequence effectively for prediction is a challenging task.

**Data Generation.** A total of 997 objects are sampled from ShapeNet, proportionally to their frequency in the dataset. Each object is normalized to fit within a bounding box of $(9\,\text{cm})^3$, and rotated so that its symmetry axis is the $x - z$ plane and it appears symmetrically to both grippers. the gripper fingers are each of size $(2\,\text{cm})^3$. For each object, 400 gripper geometries are sampled from the Bezier curve parametrization described in Sec. 5, and each gripper geometry is evaluated at increments of 0.5 cm from a height of 0.0 to until the gripper base has exceeded the height of the object. This leads to $400 * H$ gripper designs evaluated for each object, depending on its height $H$; the maximum possible number of evaluations is 7200, and the general range is from 1000 to 7200.

**Dataset Statistics.** There are a total of 4,278,400 data points in the dataset, across 997 unique objects. This evaluates to about 4290 designs per object on average, although it varies according to the range above. 31.3% of objects have a design with 6.0 reward (the highest achievable reward), 65.3% have a design with at least 5.0 reward, and 78.5% have a design with at least 4.0 reward. This indicates that most objects have a high-performing design for grasping, and the design problem is to find these high-quality designs. Only 2.7% of objects have a maximum reward of less than 1.0, a very small percentage corresponding to objects with no legitimate grasping strategy (such as a cabinet with completely flat sides, essentially a box). The dataset skews towards lower-performing designs, as is expected for a complex task of this nature. On average, a task contains 65.9% of designs with zero reward and about 4.4% of designs within 80% of the maximum reward. This creates a challenging optimization problem to find high-performing designs for a task, given that most candidates do not lead to successful grasps. Correspondingly it implies that a method that does succeed in this task is capable of addressing complex design problems.

# B   MODEL DETAILS.

Here, we present further details of our model and training setup for the gripper design task.

**Architecture of Context Encoder**. As described, $h(\mathbf{x})$ consists of three sequences corresponding to three simulations. We concatenate the information at each time-step to form one sequence as input to the context encoder. Since the sequences can have varying lengths based on when the object fell from the grasp, we pad each sequence to 5000 tokens and add a 'termination' feature for each sequence which is 0 if the sequence has not terminated and 1 otherwise. For each time-step, the two tactile images for each sequence $2 \times 16 \times 16$ are encoded by a CNN; the image encodings for each sequence are then concatenated and further concatenated to the other features for all the sequences (tactile face readings, state information). This concatenated vector is passed into an MLP which produces a final embedding for that time-step. The encoded time-steps, along with a special token for $(\mathbf{x}, f(\mathbf{x}))$ and a [CLS] token, are passed into a transformer encoder. We use sinusoidal positional encodings to add positional information for the sequence. Finally, the [CLS] token representation is added to an MLP embedding of $(\mathbf{x}, f(\mathbf{x}))$ to form the final context embedding.

**Training Setup**. As mentioned in Sec. 6, we sample context and target sets such that low-performing and high-performing designs are equally represented. In particular, since designs with 0.0 reward make up a majority of designs for most tasks, we sample context sets so that there is equal probability of zero-reward designs and nonzero-reward designs. This ensures that the context is informative for prediction and the model learns to use the tactile information in the context instead of attempting to overfit to the target set. We sample the target set by equally representing high-reward designs (at least 75% of the maximum reward), low-reward designs (0.0 reward), and 'medium'-reward designs (all others). This ensures that the model learns to predict a range of quality and not over-focus on low-performing designs, enabling it to make effective decisions for novel tasks and identify high-performing designs. We sample context size uniformly in the range [5, 30], and use a fixed target size of 100 designs.