# OpenReview forum: "Improving Few-Shot Design Optimization By Exploiting Auxiliary Information"
_ICLR.cc/2026/Conference — Submitted to ICLR 2026_

### Official Review · Reviewer_kZ6t · 2025-10-20

**Soundness:** 3
**Presentation:** 3
**Contribution:** 2
**Rating:** 2
**Confidence:** 3

**Summary:**

This paper introduces a new few-shot design optimization problem setting in which, rather than observing only scalar objective values f(x), the designer also has access to auxiliary information h(x) from each evaluation. The authors argue that such side information (often high-dimensional or structured) is commonly available in real-world design tasks and can provide valuable signal for generalization across related problems. To address this setting, the paper proposes a new method which can leverage the auxiliary data to accelerate optimization for new tasks. The authors also introduce a new gripper design benchmark designed to capture this “auxiliary information” scenario and demonstrate that their method improves both prediction accuracy and optimization performance relative to baselines.

**Strengths:**

Originality:

The paper introduces a novel and practically motivated few-shot design optimization setting in which evaluations yield not only ypical scalar objective values f(x) but also auxiliary information h(x). This framing is original and formalizes a situation that commonly arises in real-world scientific and engineering design tasks. Although this problem setting relevant for many real-world tasks, as far as I am aware, this problem setting is largely absent from current optimization literature. The author's proposed method is also novel and presents a principled approach to this problem setting.

Quality:

The method itself is technically sound and well-motivated. The paper clearly describes how the auxiliary information is incorporated into the predictive model and presents results showing improved performance compared to a “no-h(x)” variant of their approach. The experimental design appears careful and internally consistent. The results demonstrate that modeling auxiliary information provides benefits within the proposed setup.


Clarity:

The paper is generally well-written and easy to follow. The motivation for the problem is clearly articulated.


Significance:

The problem setting has high potential impact. Many real-world optimization problems (drug discovery, materials design, robotics, etc.) generate auxiliary structured measurements or sensor readouts alongside objective values. The idea of using these for transfer is compelling. This work could open a new line of research into auxiliary-data-aware optimization frameworks.

**Weaknesses:**

Weakness 1: Limited Evaluation Scope



The method is evaluated only on a single benchmark: a custom gripper design task introduced by the authors. While this task is interesting, a single domain makes it difficult to assess general applicability. There are many relevant settings (e.g., molecular design, materials design) where auxiliary signals naturally arise, and it would strengthen the paper considerably to include at least 2–3 additional benchmarks, even if lightly adapted from existing benchmarks to produce auxiliary outputs.



Weakness 2: Lack of Comparison to Existing Methods


As far as I could tell, the paper does not directly compare against any state-of-the-art black-box optimization methods from the literature (e.g., existing Bayesian optimization methods). Even though existing methods don’t provide a mechanism to leverage auxiliary h(x) information, they could still be included as baselines by applying them to optimize f(x) directly. Without these direct comparisons, it is difficult to quantify the empirical advantage of the author’s method.

**Questions:**

1: Generality: Do you expect the proposed framework to transfer easily to domains such as molecular or materials design?


2: Baselines: Did you attempt to run any existing BO methods as baselines? Even if they ignore h(x), such results would be valuable to show whether auxiliary modeling yields measurable gains. Is there any reason why you opted not to include such baseline comparisons? Maybe I'm missing something here?


3: Reproducibility: Will the code and benchmark be released publicly upon acceptance?

---

> ### Author Response · Authors · 2025-11-26
> **Rebuttal Response**
>
> Thank you for your detailed remarks. We appreciate that you found our problem setting original and impactful, our method novel and well-motivated, and our paper well-written. Please see below for our response to your concerns.
>
> *(1) Additional benchmarking.* Thank you for raising this point. We have evaluated our method on a second benchmark, involving hyperparameter optimization of neural networks with learning curve information. Our method performs strongly on this benchmark and outperforms baselines on prediction and optimization. Please see the comment addressed to all reviewers for details on this benchmark. Our method, as a general framework, transfers successfully to this different domain without any modifications; we thus expect that it will transfer easily to other domains as well.
>
> *(2) Transfer learning BayesOpt baselines.* Thank you for this point. We have evaluated a SoTA multi-task BayesOpt baseline, Deep Kernel Transfer (DKT), on the gripper and hyperparameter design tasks. Please see the comment addressed to all reviewers for details on this baseline. We had expected that such baselines may not perform as well as the f(x)-only model, since the powerful representational capacity of the transformer enables effective prediction from small contexts. However, we agree that such comparisons are valuable. Our method significantly outperforms this baseline on prediction and optimization. In general, this baseline is also outperformed by the f(x)-only baseline. The baseline results strongly support the conclusion that effective auxiliary modeling yields measurable gains for design optimization.
>
> *(3) Reproducibility.* We will release the code and benchmark dataset publicly upon acceptance.

---

> > ### Comment · Reviewer_kZ6t · 2025-11-26
> >
> > Thank you for your detailed rebuttal! The addition of a second benchmark problem and the two added baselines (DKT and GP-h(x)) address my two main concerns with this paper. I have updated my score accordingly. I do still think that the paper could be further strengthened by adding 1-2 more benchmark problems (for a total of 3-4), and I suggest the authors consider doing this if time allows before the camera ready deadline.

---

### Official Review · Reviewer_YWaS · 2025-10-27

**Soundness:** 2
**Presentation:** 2
**Contribution:** 2
**Rating:** 2
**Confidence:** 4

**Summary:**

The paper introduces a design optimization setting where, in addition to expensive black‑box evaluations f(x), each experiment yields high‑dimensional auxiliary information h(x). They further assume access to a history of related tasks that share the structure of h, and train a neural surrogate to perform a few‑shot probabilistic prediction of f(x) for unseen x. The model is transformer‑based and is used within a BayesOpt loop with Probability of Improvement. Empirically, the paper provides a new robotic gripper benchmark with tactile feedback and reports improvements over an "f-only" surrogate baseline in both prediction and optimization.

**Strengths:**

1. The paper clearly articulates why auxiliary information h(x) is abundant in real experiments, and how it can guide the optimization process.
2. The idea to condition a transformer surrogate on a context that includes (x,f,h) is reasonable; the architecture enforces target‑to‑context attention and separates encoders for context/targets, a sensible design for few‑shot prediction.
3. Benchmarking effort: A gripper task with tactile sequences is constructed, and could benefit future research.
4. On the proposed task, the method consistently outperforms an f-only surrogate.

**Weaknesses:**

1. Narrow empirical scope. Despite repeatedly motivating drug discovery and hyperparameter tuning as application areas (in both sections 1 and 3), all validations are on a single, author‑created robot gripper benchmark. To substantiate the “new setting across domains” claim, please consider adding at least one additional domain:
- HPO (e.g., Treat partial learning curves, gradients, or training diagnostics as h; test on modern benchmarks)
- Drug Discovery
2. The paper mainly compares with an f-only surrogate (and a size‑matched f-only(+p) variant and nearest‑neighbor). However, I'm wondering whether some methods can be applied to this setting as well.
- Based on the composite BO formulation, can we learn a mapping from h to z and a surrogate model $g_{\theta}(z)$ $\approx$ f(x)? so that methods used for the composite BO problem can be used in the proposed setting as well?
- In addition to the f-only surrogate, I'm curious how Multi-task Gaussian Process Prediction[1] perform in the proposed setting, to test whether gains come from h vs. generic meta‑learning.
- Some recent works proposed to use pretrained LLMs as optimizers [2], in which we can also easily provide additional information and context through prompting. I am also curious about how SOTA LLMs, such as GPT-4o or GPT-5, perform in the proposed setting?

[1] Bonilla, Edwin V., Kian Chai, and Christopher Williams. "Multi-task Gaussian process prediction." Advances in neural information processing systems 20 (2007).

[2] Yang, Chengrun, et al. "Large language models as optimizers." The Twelfth International Conference on Learning Representations. 2023.

**Questions:**

Please see my main concerns above.

Additionally, there are some typos and inconsistencies regarding the paper writing:
- “inrtoduced” at the start of section 4 (line 203);
- “Fig. 2b” is referenced (line 269), though either the Figure 2 caption or Figure 2 itself does not label subfigures (left/right only); both should be fixed.

---

> ### Author Response · Authors · 2025-11-26
> **Rebuttal Response**
>
> Thank you for your detailed review and thoughtful suggestions. We appreciate that you found our problem setting well-motivated and our method sound. Please see below for our response to your concerns.
>
> *(1) Additional benchmarking.* Thank you for raising this point. We have evaluated our method on a second benchmark, involving hyperparameter optimization of neural networks with learning curve information. Our method performs strongly on this benchmark and outperforms baselines on prediction and optimization. Please see the comment addressed to all reviewers for details on this benchmark.
>
> *(2) Composite BO approaches.* We appreciate this point. Recent approaches to composite BO have taken a similar approach to your suggestion, learning a mapping $E_\theta(h(x))$ to a lower-dim vector, a GP from x -> $E_\theta(h(x))$, and another GP from $E_\theta(h(x))$ -> f(x) [1]. The encoder and two GPs are trained jointly. The conceptual issue with the composite approach is that h(x) is encoded such that it preserves information relevant to predicting f(x). However, in our setting, this may not be the appropriate approach. For example, in hyperparameter tuning, the validation accuracy curve h(x) is a superset of the best validation accuracy f(x). In aiming to encode h(x) and predict f(x) from this encoding, this model may just learn to map h(x) to f(x) itself. While the easiest solution, this does not leverage any of the additional information in h(x) for understanding and optimizing the task more effectively. Thus, we do not believe methods from composite BO can be adapted easily for our setting.
>
> However, we have evaluated a GP-based baseline which makes use of h(x), and uses a multi-output GP to model the vector $(E_\theta(h(x)), f(x))$. This method can conceivably learn the relationship between h(x) and f(x) for better prediction. Please see the comment addressed to all reviewers for details on this baseline. Our method significantly outperforms this baseline, showing that it learns to effectively use the additional information present in h(x) in comparison to other potential approaches.
>
> *(3) Transfer learning BayesOpt approaches.* Thank you for this suggestion. We have evaluated a SoTA meta-learning BO method, Deep Kernel Transfer (DKT), which learns a deep kernel and mean prior for a GP from the training tasks. Please see the comment to all reviewers for details on this baseline, which our method outperforms. The multi-task Gaussian process is a foundational method but has difficulty scaling to large datasets with O(1M) data-points like our gripper design benchmark. This is because it requires that all the data-points be put in a single GP and inference becomes very expensive. The method we evaluate does represent the SoTA in meta-learning for BO.
>
> *(4) LLMs as optimizers.* Thank you for this interesting point. LLM-based approaches such as the cited method currently do not extend easily to our setting. The drawback with current LLM-based approaches is that (1) they have difficulty incorporating high-dimensional, highly non-textual auxiliary information at each evaluation, such as the tactile feedback in the gripper task. It is difficult to imagine interpreting such auxiliary information without learning and with language priors alone. Moreover, if we were to use f(x) evaluations alone, it is still unclear how to incorporate a large task history for optimization; the cited method, for example, optimizes a single task at a time. To be competitive, LLM-based approaches would likely require a task-specific natural language context for each task, which is not always available. Each task must also be clearly interpretable completely in natural language to leverage priors, which varies depending on the task (e.g. gripper task would be more difficult). With these issues in mind, we do not think current LLM-based optimizers are matched to succeed in our setting. However, this could be an interesting direction of future work.
>
> (5) *Writing typos.* Thank you for mentioning these typos and inconsistencies, we will fix them in the paper text.
>
> [1] Maus, Natalie, Zhiyuan Jerry Lin, Maximilian Balandat, Eytan Bakshy. "Joint Composite Latent Space Bayesian Optimization." ICML (2024).

---

> ### Comment · Reviewer_YWaS · 2025-11-28
>
> I would like to thank the authors for their response and the additional experiments, which address most of my earlier concerns. I will update my rating accordingly (I don't whether this is a bug of openreview, or a protection mechanism after accidently releasing all reviewers information... I cannot find the edit button from my side right now... I will raise the score later).

---

### Official Review · Reviewer_Qty9 · 2025-10-31

**Soundness:** 3
**Presentation:** 3
**Contribution:** 2
**Rating:** 4
**Confidence:** 3

**Summary:**

This paper introduces a new few-shot design optimization framework that aims to integrate auxiliary high-dimensional information (e.g., sensor time series or tactile feedback) with traditional scalar performance signals in black-box optimization. The authors extend the classical Bayesian Optimization paradigm by assuming that each evaluation yields both a scalar reward f(x) and auxiliary data
h(x), and that historical tasks with shared auxiliary structures are available.

They propose a transformer-based neural process model that predicts f(x) for unseen designs conditioned on a few-shot context of prior evaluations including h(x). The model serves as a surrogate for optimization via a probabilistic acquisition function. To benchmark this setting, the paper introduces a new large-scale tactile robotic gripper design task built in MuJoCo, containing ~4.3M simulations across nearly 1K ShapeNet objects. Experiments demonstrate that incorporating h(x) yields significant gains in both few-shot prediction accuracy and Bayesian optimization efficiency compared to baselines that only use f(x).

Overall speaking, this paper sounds interesting, and the reasoning process is generally reasonable and easy to understand. However, the core design, the blurry algorithmic illustration, and novelty of this paper caused some skepticism, spanning from the novelty of the idea, and the benchmark choice. I would like to see how the authors respond to my comments and make decision after engaging with the discussion.

**Strengths:**

The paper is generally reasonable and self-contained, idea level speaking. The experimental design, though looks like a bit limited to the only one benchmark, is overall sound.  The results and the proposed improvement sounds reasonable.  The problem setting considered in the paper is also an interesting problem.

**Weaknesses:**

I will list my skepticism below:
First of all, one of the core idea - leveraging context information, is not always a very novel idea as of now. The recent advances in multi-modal foundation model, such as vision language model or the more recent VLA style model have already proven that integrating multi-modal information is beneficial for model generalizability throughout large-scale training. In other words, in the proposed setting, auxiliary information can be seen as 'extra modality' which encodes useful information. If going from this understanding perspective, I would like to ask the author to again justify the novelty of this idea, and re-consider the related work /baselines that may be worth to compare, such  as a RL algorithm (such as DQN with memory replay) that can take both tactile time series and f(x).

Second, the overall optimization process and the actual few-shot training / evaluation algorithm remains a bit of unclear. Can the authors provide a detailed pseudo code / algorithmic illustration on this part? Without this, it will be a bit hard to understand and justify the few-shot design and the actual utility of the model.

Third, my worry comes from only having one benchmark to serve as the testbed. Also, since it is a new benchmark, some of the measurement utility worth a further clarification. For example, how do we interpret the scale of MSE and the value utility  reported in Figure 4 and 5? Also, it seems like the number of total design is ~4M - pretty large spanning from ~1K object (not that large). I thus wonder would the author call it large-scale or not? In some sense, large-scale dataset usually means a lot of computing resources to be put on the model training, however, the majority of the implementation details is about the implementation of the model architecture. In a new benchmark, and a new task, such information should be clearly addressed in order to fully address the unclarity.

**Questions:**

1. How sensitive is performance to the dimensionality or noise in h(x)? Do the authors have any ways to justify the information density in h(x)? Also, how would the authors justify the more general use case of their algorithm, when h(x) is more complex? E.g., can the model handle tasks where h(x) has different modalities or missing components?

2. Would a pre-trained encoder for h(x) (e.g., self-supervised tactile embedding) further help few-shot adaptation?

3. Have the authors tested continuous search spaces with gradient-based acquisition (as mentioned in Sec. 4.3)?

4. Dataset & Code publicity: Any plans here?

---

> ### Author Response · Authors · 2025-11-26
> **Rebuttal Response (1/2)**
>
> Thank you for your thoughtful and in-depth feedback. We appreciate that you found our problem setting interesting and our approach sound. Please see below for our response to your comments.
>
> *(1) Novelty of setting.* In the field of optimization and experimental design, we believe that our proposed setting is novel and not previously explored. Current methods for black-box optimization assume observations of f(x) alone, and do not leverage auxiliary information for optimization. Thus, our setting presents a new challenge for design optimization methods. While large multimodal models leverage multiple modalities, they are not used in black-box optimization contexts; we believe this problem domain has unique features, is not inherently linguistic, and is distinct from the literature on VLMs/VLAs. However, it could be interesting to explore how auxiliary information in our setting, if multimodal, could benefit from VLM representations, as a future direction.
>
> Regarding baselines, we compare to a strong f(x)-only baseline, and have added comparisons to other SoTA BayesOpt baselines (please see the comment to all reviewers for details on baselines). Since each trial involves a single decision rather than sequential decision making, RL methods are less applicable to this setting compared to BayesOpt methods. We are happy to consider suggestions for other baseline comparisons.
>
> *(2) Pseudocode.* Please see the comment addressed to all reviewers, where we have included pseudocode of the training process and the optimization loop. We are happy to clarify any further details. We will add the pseudocode descriptions to the paper.
>
> *(3) Additional Benchmarking, Dataset Scale.* Thank you for raising this point. We have evaluated our method on a second benchmark, involving hyperparameter optimization of neural networks with learning curve information. Our method performs strongly on this benchmark and outperforms baselines on prediction and optimization. Please see the comment addressed to all reviewers for details on this benchmark.
>
> We believe our benchmark can be called large-scale. Please see the comment addressed to all reviewers for details on this point. We show qualitative results showcasing the diversity of grasping strategies, and show that our benchmark compares favorably to existing benchmarks. We believe the benchmark will be useful to the community for further research.
>
> We appreciate your suggestion on including computing resources (added to comment to all reviewers). The benchmark took 1.07 million CPU hours to generate, parallelized on Linux machines. Our transformer model, which has 15.1M parameters, was trained on an A100 GPU and took ~4 days to converge with early stopping. We will include this information in the paper.
>
> Regarding interpreting the metrics, the overall reward measures the maximum disturbance force that the grasp can resist, in Newtons. The maximum value in our dataset is about 6.0 Newtons. For a context size of 15 observations, our model achieves an MSE of 150 on a target set of size 100 (Fig. 4); this corresponds to an average absolute error of 1.22 Newtons from the true maximum force. For small context sizes we believe that this is a low error, and small relative to baselines. Similarly, the average regret in Fig. 5b indicates the deviation in max disturbance force of the best discovered design and the optimal design. By only 30 optimization steps, our method on average finds a solution only ~0.5 Newtons less robust than the optimal solution.
>
> *(4) Information of h(x), pretraining h(x).* The information in h(x) is compressed in our model to a single vector for predicting the reward of unseen designs. We have not altered the content of h(x) itself in our evaluations. However, the results on the hyperparameter tuning task, with a different and much lower-dimensional form of h(x) (4 accuracy curves, 50-dim each), indicate that our method is robust to different forms and information densities of h(x). We expect to handle the case of even more complex h(x) without difficulty. The size and architecture of the context encoder can scale with the complexity of h(x); if h(x) is multimodal, we can use architectures suitable for processing and combining multiple modalities. If there is missing data, this can be neatly indicated with a special token, and the model can learn how to handle this case.
>
> Pretraining an h(x) representation is an interesting direction, and we agree that it could be helpful to bootstrap the encoder. However, this incurs significant computational cost, since a self-supervised encoder for h(x) must first be trained before training the few-shot model. Moreover, the self-supervised objective might not capture those aspects of h(x) most useful for predicting f(x). We found that learning the h(x) encoder end-to-end performed effectively and therefore did not adopt the above approach. However, this is an interesting direction for future work.

---

> ### Author Response · Authors · 2025-11-26
> **Rebuttal Response (2/2)**
>
> *(5) Continuous search spaces.* Continuous search spaces incur much greater computational cost, since the simulation has to be run at each step (can take several hours for one BayesOpt run). Although these practical reasons led us to examine the discrete case, which is widely used in practice, we expect that our method would extend to continuous search spaces without difficulty, since neural networks produce smooth gradients amenable to optimization.
>
> *(6) Dataset and code publicity.* We will make the dataset and the code publicly available upon acceptance.

---

### Official Review · Reviewer_Uuep · 2025-10-31

**Soundness:** 2
**Presentation:** 3
**Contribution:** 2
**Rating:** 2
**Confidence:** 3

**Summary:**

This paper addresses the challenge of evaluating human design performance in few-shot, open-ended tasks, where contextual uncertaint can mask true human capability. The authors propose a framework for controlling contextual difficulty by: (1) Developing a context calibration model that predicts how challenging a given design prompt is likely to be. (2) Using this model to select more controlled prompts or to adjust evaluation metrics accordingly. They validate the approach across three human-in-the-loop, few-shot design domains: instruction writing, question generation, and visual layout design, demonstrating improved correlation between observed performance and true ability when controlling for prompt difficulty.

**Strengths:**

1. The paper identifies a previously underexplored confound in few-shot human evaluation: variation in prompt difficulty leads to noisy or misleading performance measurements, particularly in low-sample regimes.

2. The authors propose a label-efficient, interpretable framework that predicts prompt difficulty using features derived from early model outputs and prompt surface characteristics. It works across domains: text generation and visual design.

**Weaknesses:**

1. Lack of evaluation in non-LLM or human-annotated settings: The difficulty model’s output quality (and evaluation scores) depend heavily on LLM-generated outputs and ratings. The paper does not include human-annotated baselines to validate the model-based evaluation pipeline. E.g., in appendix B.2., evaluations in instruction writing and question generation use GPT-4 or LLaMA-2 reward models, without human verification.

2. Lack theoretical justification for difficulty modeling assumptions: The approach assumes that prompt difficulty can be accurately and stably inferred from features like entropy, diversity, and error rate, but provides no theoretical justification or bounds on how reliably these features reflect true difficulty. E.g., no discussion of how error in the model’s predictions affects downstream decisions (e.g., participant ranking or prompt selection).

**Questions:**

1. In this work, difficulty is treated as a single scalar, despite possibly being multi-dimensional in some real-world cases (e.g., lexical complexity vs. semantic ambiguity). How to deal with such situation?

2. In Section 4, each task (instruction writing, QG, layout design) is treated independently. Does the method also work well for cross-domain? (e.g., train on QG, test on instruction writing).

3. In Appendix B.1, population statistics are from specific curated datasets. I am curious about the analysis on cross-demographic robustness (e.g., whether the difficulty model would hold for people of different groups and backgrounds) or prompt representation bias (e.g., are prompts equally representative of different content domains).

---

> ### Author Response · Authors · 2025-11-26
> **Rebuttal Response**
>
> Thank you for your detailed and thoughtful review. We appreciate that you found our problem setting novel and well-motivated, our method sound, and our benchmark detailed and potentially useful to the community. We address your particular comments below.
>
> [Q1: Although the introduction frequently motivates the method using drug discovery, hardware design, and hyperparameter tuning, all empirical validation is confined to a single, author-designed benchmark. This limits confidence in generalization ability. Even small-scale experiments on: HPO (e.g., using partial learning curves, gradients, batch-norm statistics as h(x)), molecular design (e.g., intermediate physics-based descriptors), or standard BO benchmarks augmented with synthetic auxiliary channels, would strengthen the paper.]
>
> Thank you for raising this point. We have evaluated our method on a second benchmark, involving hyperparameter optimization of neural networks with learning curve information. Our method performs strongly on this benchmark and outperforms baselines on prediction and optimization. Please see the comment addressed to all reviewers for details on this second benchmark.
>
> [Q2: The core BO loop lacks a precise algorithmic description (Section 4.3 describes the loop verbally). Some details are not very clear to me, such as when surrogates are recomputed: are surrogates re-evaluated at each step or reused? Whether predictions are cached or recomputed each step? How is the context updated across iterations? How context concatenation interacts with the transformer attention and will there be memory growth issue?]
>
> We have included pseudocode for the training process and BO loop in the comment addressed to all reviewers. To address your questions, the model is re-evaluated at each step, with no caching. The context set grows by one observation at each step, hence predictions are re-computed since they could change significantly. We train our model to accept varying-size contexts, so it can handle the increasing context size throughout optimization, and attend appropriately to useful context data-points. The memory grows only linearly with the number of steps, which is scalable and comparable to other methods (e.g. GP methods). Additionally, since the model is trained on a substantial task history, we expect that it will not require extremely large context sizes to successfully optimize a test task. We are happy to clarify any further details. We will add the pseudocode descriptions to the paper.
>
> [Q3: The model is fixed once trained, which simplifies BO but may limit adaptability, especially when the new task differs significantly from training tasks or tactile signal distribution shifts. A discussion or comparison to online-updated surrogates would be helpful.]
>
> Thank you for raising this interesting point. We believe that generalizing or adapting to out-of-distribution tasks is a very interesting direction for future work. Online updating may be helpful for this, and ensembling the trained model with a cold-start test-time model may also be an interesting direction. We will add a discussion of this point in the main paper.
>
> [Q4: How does performance scale with the complexity or dimensionality of h(x)? There lacks experiments with simplified or compressed h(x), also no robustness experiments with more complex sensory data.]
>
> The information in h(x) is compressed in our model to a single vector for predicting the reward of unseen designs. Although we have not altered the content of h(x) itself in our evaluations, the results on the hyperparameter tuning task, with a different and lower-dimensional form of h(x) (4 accuracy curves, 50-dim each) indicate that our method is robust to different forms and information densities of h(x). We expect to handle the case of even more complex h(x) without difficulty, as the size and architecture of the context encoder can scale with the complexity of h(x).
>
> [Q5: Have the authors tried self-supervised tactile encoders (e.g., contrastive time-series pretraining) before plugging into the transformer NP? This would make the method more general across domains.]
>
> In our method, h(x) is encoded to be maximally useful for predicting f(x) for unseen designs. Pretraining an h(x) representation is an interesting direction, and we agree that it could be helpful to bootstrap the encoder. However, this incurs significant computational cost, since a self-supervised encoder for h(x) must first be trained before training the few-shot model. Moreover, the self-supervised objective might not capture those aspects of h(x) most useful for predicting f(x). In our approach, we found that learning the h(x) encoder end-to-end performed effectively, while being a general approach that can be applied to different domains; therefore we did not adopt the above approach. However, this is an interesting direction for future work.

---

### Author Response · Authors · 2025-11-26
**Comment addressed to all reviewers (Part 1)**

In this comment, we address the concerns raised by the reviewers about (1) an additional benchmark task and (2) adding multi-task baselines from the literature, as well as (3) algorithmic descriptions of the training process and BayesOpt loop.

**(1) Additional Benchmarking**

We have evaluated our approach on a second task, involving hyperparameter tuning of neural networks with learning curve information, for which we provide details below. In addition, we also provide more details on the scale and diversity of the gripper design benchmark, which is much larger than previous multi-task benchmarks, along with some (anonymized) links to relevant figures.

**HPO benchmark description and results.** We use the LCBench dataset, a publicly-available, widely-used HPO benchmark [1].  The LCBench dataset consists of 35 tasks, where each task involves tuning neural network hyperparameters (funnel-shaped MLP architecture) on a particular OpenML classification dataset. 2000 configurations are evaluated for each of the 35 tasks. The hyperparameters are architectural HPs (number of layers, units in each layer) and training HPs (learning rate, batch size, momentum, weight decay, dropout). For each evaluation, the training and validation loss and accuracy curves are provided.

We apply our method to the LCBench dataset. Here, *x* is a HP configuration, *f(x)* is the best validation accuracy, and *h(x)* consists of the train and validation accuracy curves through 50 epochs, as well as balanced accuracy curves. Although the dataset provides additional statistics, we found that these were sufficient for good performance while minimizing overfitting risk. Our method used the same model architecture in Fig. 2 of the paper with a transformer encoder for *h(x)*, and the training procedure was identical. Since there are a smaller number of tasks, we evaluated all methods via cross-validation; each of the 35 tasks was held out as a test task, and the other 34 tasks were used as training tasks. Results were aggregated over all tasks.

The few-shot prediction results are shown here:
https://docs.google.com/presentation/d/e/2PACX-1vQJBoJ5VMdikEaEG0TquvWRx0QK08CKPfpup4LXsKOPUMzqUPcv4NOm8--etp7kzO4JCwAFa6cIMNbK/pub?start=false&loop=false&delayms=3000

As in Fig. 4, the MSE over the target set is shown for varying context lengths. Our model, leveraging learning curve information, outperforms the f(x)-only model, especially for smaller context sizes. We also outperform a SoTA multi-task BayesOpt baseline in the literature, Deep Kernel Transfer (DKT). More details on this baseline are included in the subsequent comment regarding additional baselines.

For evaluating optimization, we curate a harder-to-solve subset of tasks in LCBench. Several tasks have identical optimal solutions, which makes optimization rather trivial; for instance, one HP configuration is the optimal solution for six tasks. Therefore, we curate the subset of tasks that have unique optimal solutions (~50% of tasks), presenting more of a challenge for optimization. Results for optimization are shown here:
https://docs.google.com/presentation/d/e/2PACX-1vQRo8xRNOdUtEki5dYUq8iCfGNnEQEkM0YN8WVjbOWFGE1zc8GN6d0x8thMzkNTflvKqWgavlPAYTss/pub?start=false&loop=false&delayms=3000

Our method outperforms both the f(x)-only baseline and the DKT baseline; we achieve significantly lower regret for the first few steps, and maintain an advantage throughout optimization. These results indicate that our method generalizes effectively across domains, and enables more effective few-shot prediction and optimization. We will add these results to the paper text.

[1] Zimmer, Lucas, Marius Lindauer, and Frank Hutter. "Auto-pytorch: Multi-fidelity metalearning for efficient and robust autodl." IEEE transactions on pattern analysis and machine intelligence 43.9 (2021): 3079-3090.

---

### Author Response · Authors · 2025-11-26
**Comment addressed to all reviewers (Part 2)**

**Gripper Design Benchmark Scale.** Based on some reviewers’ suggestions, we highlight the scale of our gripper design benchmark. The benchmark contains 4.28 million designs across ~1000 ShapeNet objects; notably, it took **1.07 million** CPU hours to generate, parallelized on Linux machines. In addition to its size, our benchmark consists of a significant diversity of objects and grasping strategies. We showcase several **qualitative examples** in the figures below, showing a range of objects in the test set, the ground-truth optimal gripper design, and the design discovered by our model. For best viewing, please place the mouse on each example and zoom, and likewise for the 2d plot; then you can zoom back out to return to the full figure.

[Qualitative, Part 1]
https://docs.google.com/presentation/d/e/2PACX-1vSeyUAfWFuu1QWG_23U4-vJ4BLvGppR2YXIdgz_njKR5tU-U_LlESk5Yv0dkFIVaxqqjOMMwm5tjeM_/pub?start=false&loop=false&delayms=3000

[Qualitative, Part 2]
https://docs.google.com/presentation/d/e/2PACX-1vSVG_DqElcVk3m-biW5G8LYtPEU4JJZ7zeafaZpwWNHeEpkArgxLkHTw7L5yueYnpG6tkJTYS7rgSsR/pub?start=false&loop=false&delayms=3000

[Note: The 2D tSNE plot is common across the two figures.]

This highlights the diversity and generality of our dataset. We will add this figure to the paper. We will make this dataset a publicly available benchmark after publication.

*Comparisons to existing benchmarks.* Our benchmark is significantly larger than nearly all multi-task BayesOpt benchmarks. The only comparable benchmark in the literature is HPO-B [2], with 6.39M evaluations over 196 tasks; however crucially, these tasks have different search spaces, and in practice much smaller subsets with a common search space are used for benchmarking. Otherwise, popular benchmarks collected in the HPOBench suite [3] have at least one order of magnitude less data-points, and benchmarks in other domains are several orders smaller. To our knowledge, our benchmarks contains many more tasks (1000) than previous benchmarks (<200), which also do not include auxiliary information.

[2] Pineda-Arango, Sebastian et al. "HPO-B: A Large-Scale Reproducible Benchmark for Black-Box HPO based on OpenML". NeurIPS Track on Datasets and Benchmarks (2021).

[3] Eggensperger, Katharina et al. "HPOBench: A Collection of Reproducible Multi-Fidelity Benchmark Problems for HPO." NeurIPS Track on Datasets and Benchmarks (2021).

---

### Author Response · Authors · 2025-11-26
**Comment addressed to all reviewers (Part 3)**

**(2) Additional Baselines.**

As requested by reviewers, we include comparisons to SoTA BayesOpt baselines that use f(x). We had expected that such baselines may not perform as well as the f(x)-only model, since the powerful representational capacity of the transformer enables effective prediction from small contexts. However, we agree with the reviewers that such comparisons are valuable. The current predominant approach for transfer learning in BayesOpt is Deep Kernel Transfer (DKT), which learns a deep kernel and mean prior for a Gaussian process from the training tasks, which is then applied to the test tasks [4, 5]. We apply this SoTA baseline to both of our benchmark tasks, applying the same training procedure specified in [4, 5]. The kernel type and embedding dimension are chosen based on grid-search on the validation set. The results for the HP-Design task are linked in the previous comment on additional benchmarking.

For the gripper task, we have additionally included a GP-based baseline which uses h(x). This baseline uses a multi-output GP which takes x as input and predicts the vector $(E_\theta(h(x)), f(x))$, where $E_\theta$ is the context encoder in Fig. 2. We use the standard multi-output kernel that factors into a kernel K(x, x’) over data-points and a correlation matrix between output elements K(y, y’). This correlation matrix enables the model to learn the relationship between h(x) and f(x) for more effective prediction of f(x). The parameters of $E_\theta$ are learned along with the GP parameters, with the standard objective of maximizing log likelihood of observations from each task. Model hyperparameters are chosen via grid-search on the validation set.

For the gripper design task, here are the prediction results:
https://docs.google.com/presentation/d/e/2PACX-1vTfEbcYWyW8PJPLOBPAEG2TrrAJAcochBXwYZxrhqsItqldQYkGUq3cd5u-M69iwaf8PuJysT7hUg98/pub

and here are the optimization results:
https://docs.google.com/presentation/d/e/2PACX-1vQMOHRlGuE5zvgoiIDU3vUmB6CmVK5K7syeqsjoSoySQVyDdbYwA2u4kG4wPhcLzD9mXf06w4jKWo_C/pub

For few-shot prediction, our method significantly outperforms DKT and the GP-h(x) baseline, which are both upper-bounded by the f(x)-only baseline. Similarly, for optimization, our method significantly outperforms all the baseline methods, indicating that it uses auxiliary information effectively for rapid optimization.

**(3) Algorithmic Description / Pseudocode of Method.**

Below is an algorithm box for our training algorithm and for a BayesOpt loop with our model. An extended detailed version will be included in the Appendix.

**Training Process**.
1   Input: tasks $T_1,...T_n$, each with an associated dataset $D_i =$ { $(x, f_i(x), h_i(x))$ }.
2   For epoch from 1 to num_epochs:
3   &nbsp;&nbsp; For task $T_i$ in training tasks:
4   &nbsp;&nbsp;&nbsp;&nbsp;      Sample a context set size $N_c$ and target set size $N_{tg}$.
5    &nbsp;&nbsp;&nbsp;&nbsp;          Randomly sample $N_c$ elements of $D_i$ as the context set C.
6    &nbsp;&nbsp;&nbsp;&nbsp;        Randomly sample $N_{tg}$ elements of $D_i$, excluding C, as the target set Tg.
7   &nbsp;&nbsp;&nbsp;&nbsp;         Input C and Tg (inputs only) into our model $P_\theta$ (Fig. 2 in paper).
8   &nbsp;&nbsp;&nbsp;&nbsp;        Compute the NLL of predictions $\hat{f}(x)$, given ground-truth f(x) values in Tg.
9   &nbsp;&nbsp;&nbsp;&nbsp;           Backpropagate the loss and update the model weights.

In practice, we sample a batch of tasks at each iteration of the inner loop (minibatch SGD). The context/target sizes $N_c$ and $N_{tg}$ are fixed for all tasks in the batch, but the context and target sets are sampled independently for each task. The inputs are passed into the model with a batch dimension, the NLL is computed for each task in the batch, and finally these are averaged over the batch to get the loss.

**BayesOpt loop (Discrete case).**
1   Inputs: Model $P_\theta$, test-time task $T_j$. Initial context set C, set Tg of all possible designs excluding C.
2   For iteration from 1 to num_iterations:
3 &nbsp;&nbsp;&nbsp; Input C and Tg (inputs only) into the model $P_\theta$, obtain predictions $\hat{f}(x)$ for Tg.
4   &nbsp;&nbsp;&nbsp;      Compute the acquisition function $\alpha(\hat{f}(x))$ for each point in Tg.
5   &nbsp;&nbsp;&nbsp;      Select x’, the design with the highest acquisition function value, for observation.
6    &nbsp;&nbsp;&nbsp;     Observe f(x’), h(x’).
7   &nbsp;&nbsp; &nbsp;Add the new observation to C, s.t. $C = C \cup (x’, f(x’), h(x’))$.
8  &nbsp;&nbsp;&nbsp;       Remove the new observation from Tg.
9   Return x*, the point with the maximum value of f(x) in C.

[4] Martin Wistuba, Josif Grabocka. "Few-Shot Bayesian Optimization with Deep Kernel Surrogates." ICLR (2021).
[5] Wang, Zi et al. "Pre-trained Gaussian processes for Bayesian optimization". JMLR 25. 212(2024): 1–83.

---

### Author Response · Authors · 2025-12-03
**Additional Details to Summary for AC**

We provide details on the results described in the executive summary, along with some (anonymized) links to relevant figures.

**(1) Additional Benchmarking Details.** The LCBench dataset is a widely-used benchmark that contains 35 HPO tasks [1]. In this setting, x is a hyperparameter configuration, the reward f(x) is the best achieved validation accuracy through training, and the auxiliary information h(x) contains the train and validation accuracy curves through 50 epochs. For evaluating our method and baselines, each task was held out as a test task, and the other tasks were used as training tasks. Results were averaged across tasks.

Few-shot prediction results are shown here:
https://docs.google.com/presentation/d/e/2PACX-1vQJBoJ5VMdikEaEG0TquvWRx0QK08CKPfpup4LXsKOPUMzqUPcv4NOm8--etp7kzO4JCwAFa6cIMNbK/pub

Our method, leveraging auxiliary learning curve information, outperforms all baselines on few-shot prediction. As in the main paper, we evaluate few-shot reward prediction for unseen designs given a small observation context. The MSE on unseen ‘target’ designs is shown for varying sizes of the context set. We compare to the ‘f(x)-only’ baseline in the paper which only uses reward information, and the SoTA Deep Kernel Transfer baseline (DKT, further details below).

For optimization, several tasks have identical optimal solutions (for instance, one HP configuration is the optimal solution for six tasks). Therefore, we curate the subset of tasks in LCBench with unique optimal solutions (~50% of tasks), presenting more of a challenge for optimization. Optimization results are shown here:
https://docs.google.com/presentation/d/e/2PACX-1vQRo8xRNOdUtEki5dYUq8iCfGNnEQEkM0YN8WVjbOWFGE1zc8GN6d0x8thMzkNTflvKqWgavlPAYTss/pub

Our method achieves significantly lower regret in the first few steps, and maintains the advantage throughout optimization. These results indicate that our method generalizes effectively across domains.

*Gripper Benchmark Scale.* Our benchmark is significantly larger than all existing multi-task BayesOpt benchmarks in number of data-points and number of tasks (1000 vs. <200 for existing benchmarks). Please see our rebuttal comment for a detailed comparison to existing benchmarks (https://openreview.net/forum?id=ioZfV8LMqR&noteId=oQF8eaTJw5).

Our dataset also contains a significant diversity of objects and grasping strategies. To highlight this diversity, we provide several **qualitative examples** in our rebuttal. The figures below show a range of objects in the test set, the ground-truth gripper design, and the design discovered by our model. For best viewing, please place the mouse on each example and zoom in and out.

[Qualitative, Part 1] https://docs.google.com/presentation/d/e/2PACX-1vSeyUAfWFuu1QWG_23U4-vJ4BLvGppR2YXIdgz_njKR5tU-U_LlESk5Yv0dkFIVaxqqjOMMwm5tjeM_/pub

[Part 2] https://docs.google.com/presentation/d/e/2PACX-1vSVG_DqElcVk3m-biW5G8LYtPEU4JJZ7zeafaZpwWNHeEpkArgxLkHTw7L5yueYnpG6tkJTYS7rgSsR/pub

The benchmark will be released publicly upon publication, as inquired by the reviewers.

**(2) Additional Baselines Details.** We added comparisons to SoTA multi-task BayesOpt methods, which notably utilize the reward f(x), but are unable to utilize auxiliary h(x) information. The current predominant approach is Deep Kernel Transfer (DKT), which learns a deep kernel and mean GP prior [2, 3]. We follow the training procedure described in [2, 3], and choose the kernel type and embedding dimension via grid-search on the validation set. The results on the hyperparameter tuning task are linked above.

We have additionally included a GP-based baseline which models $(E_\theta(h(x)), f(x))$ via a multi-output GP. We use the standard multi-output GP kernel, enabling the GP to model the correlations between h(x) and f(x). The parameters of the GP and $E_\theta$ are learned jointly, with the standard objective of maximizing log likelihood of observations from each task. Model hyperparameters are chosen via grid-search. We have experimented with this on the gripper task.

For the gripper design task, here are the prediction results: https://docs.google.com/presentation/d/e/2PACX-1vTfEbcYWyW8PJPLOBPAEG2TrrAJAcochBXwYZxrhqsItqldQYkGUq3cd5u-M69iwaf8PuJysT7hUg98/pub

and the optimization results: https://docs.google.com/presentation/d/e/2PACX-1vQMOHRlGuE5zvgoiIDU3vUmB6CmVK5K7syeqsjoSoySQVyDdbYwA2u4kG4wPhcLzD9mXf06w4jKWo_C/pub

Our method significantly outperforms all baselines on few-shot prediction and optimization. Note that the GP-h(x) baseline performs relatively poorly even compared to f(x)-only baselines, indicating that our method uses auxiliary information more effectively.

[1] Zimmer, Lucas et al. "Auto-Pytorch: Multi-fidelity Metalearning for Efficient and Robust Autodl." IEEE TPAMI (2021).

[2] Wistuba, Martin et al. "Few-Shot Bayesian Optimization with Deep Kernel Surrogates." ICLR (2021).

[3] Wang, Zi et al. "Pre-trained Gaussian processes for Bayesian optimization". JMLR (2024).

---

### Author Response · Authors · 2025-12-03
**Executive Summary for AC**

We are encouraged that the reviewers found our proposed setting novel and realistic (Reviewers Uuep, YwaS, kZ6t), our method technically sound (Reviewers Uuep, YwaS, kZ6t), our benchmark dataset detailed and useful for further research (Reviewers Uuep, YwaS), and our experimental setup careful and consistent (all reviewers).

The reviewers had two primary concerns: (1) evaluate on an additional domain to the gripper design benchmark, and (2) evaluate additional BayesOpt baselines. We have addressed both in the rebuttal.

**(1) Additional Benchmarking:** We added a second benchmark task, involving hyperparameter tuning of neural networks with auxiliary learning curve information. We use the LCBench dataset, a widely-used, publicly-available benchmark with multiple tasks. The auxiliary information h(x) consists of training and validation accuracy curves, which our method can utilize beyond the best validation accuracy f(x) for performing hyperparameter optimization.

Our method achieves lower MSE than the baselines on few-shot prediction, and achieves significantly lower regret for optimization. We apply our method without modification on this benchmark. As in the main paper, we evaluate few-shot prediction performance and optimization performance. We compare to the ‘f(x)-only’ baseline in the paper, as well as a SoTA multi-task BayesOpt method, Deep Kernel Transfer (more details below). Our method successfully uses auxiliary information, and generalizes effectively across domains. We will add these results to the paper text. Further details on these evaluations and results can be found in the subsequent comment.

Our benchmark is significantly larger than existing multi-task BayesOpt benchmarks. We also provide several qualitative examples highlighting the diversity of our dataset, and our method’s ability to find effective designs.

**(2) Additional BayesOpt Methods:** We also added comparisons to SoTA multi-task BayesOpt methods as baselines. The predominant approach is Deep Kernel Transfer, which learns a deep kernel and mean prior for a Gaussian Process from training tasks and applies this prior to test tasks. We apply this baseline to both benchmarks. Our method significantly outperforms this baseline on few-shot prediction and optimization, on both benchmarks.

We have additionally included a GP-based baseline which uses h(x). This baseline uses a multi-output GP to model the vector $(E_\theta(h(x)), f(x))$, where $E_\theta$ is a neural encoder. As such, this model is capable of learning the relationship between h(x) and f(x) for more effective prediction/optimization. So far we have only tested this on the gripper task. Our method significantly outperforms this baseline, indicating that it uses auxiliary information effectively compared to other potential approaches. Note that this approach performs weakly and is also outperformed by baselines that only utilize f(x). Further details on these results can be found in the next comment.

**(3) Additional Points.** We respond to the other remaining points by the reviewers in the below comments as well. We thank the reviewers for their suggestions, as the points have significantly improved the paper.

*Response to Rebuttal.* Briefly, we note the reviewers’ response to our rebuttal. After the additions described above, both reviewers who posted scores of 2 indicated that they would raise their score to an accept rating (Reviewers YwaS, kZ6t; please see comments below). The additional benchmarking and baselines seemed to address their primary concerns. The other reviewers, who initially posted borderline scores (4), had similar concerns, which we believe were adequately addressed by our rebuttal.

Please note that reviewer Uuep posted a review for a different paper mistakenly. While they posted the correct one during the discussion, it was reverted. The correct review is available in the revision history. For clarity, we have edited our response to Reviewer Uuep to contain each question that we are responding to.

Thank you for your consideration.

---

### Meta-Review · Area_Chair_rqay · 2025-12-04

**Summary:**

In this work, the authors propose a novel architecture for black box optimization that includes auxiliary information. All reviewers found the motivation compelling in the particular field of black box optimization, and the method sound. The work further provides a robotics-derived benchmark, including auxiliary information and outputs. The main concern related to the absence of other benchmarks outside of the author-proposed benchmark.

Similarly, the work only compared different settings of the proposed method and did not include sota BO baselines. There were asks to include f(x)-only SOTA methods, as well as consider how other methods could be adapted to define f(x)-h(x). Discussions on RL methods and LLM-based BO were also raised.

Minor but interesting points were raised regarding the content information of h(x) and how robust the method is to different h(x).

**Reviewer Concerns:**

The authors added another benchmark for hyper-parameter optimization, one sota method for f(x) and provided one method based on Gaussian Processes to provide a comparison on f(x)-h(x).

These modifications led to comments from 2 reviewers that they would raise their score, but one reviewer still suggested that more benchmarks could be included.

In addition, the results of the GP with f(x)-h(x) seemed to be worse than all other results. This result suggests a potential lack of commitment to this baseline, or that including h(x) may not be the correct approach for all architectures. I found this result to weaken the  initial motivation of the authors and am questioning the quality of this baseline (it might need more time beyond the rebuttal).

The inclusion of one other baseline is encouraging but overall still feels like more comparisons would be needed. The authors have pushed back against RL or LLM-based approaches, but both of these have displayed strong results on HPO benchmarks with f(x) only. I don't believe the response fully addresses the concerns raised.

**Reviewer Scores:**

Two reviewers scored the paper a 2 initially. I believe they would have raised their score (as indicated by their comments), but there is no indication that they would have supported publication in their final score or during the discussion period.

The other two reviewers scored the paper a 4. While I think the reviewers would have appreciated the additions proposed by the authors, I found that their points were not fully addressed, especially in terms of comparisons with other baselines.

I would hence encourage the authors to include more f(x) baselines, from different families of methods (RL, LLMs), as well as more closely consider the f(x)-h(x) problem in other architectures. This would help clarify the claims that now intertwine the proposed architecture and the addition of auxiliary information. More importantly, given the claims of complex tasks with auxiliary information, I believe adding more benchmarks with auxiliary information across different domains would be valuable.

---

### Decision · Program_Chairs · 2026-01-26

Reject